# Modulation of Early Neutrophil Granulation: The Circulating Tumor Cell-Extravesicular Connection in Pancreatic Ductal Adenocarcinoma

**DOI:** 10.3390/cancers13112727

**Published:** 2021-05-31

**Authors:** Harrys Kishore Charles Jacob, John Lalith Charles Richard, Rossana Signorelli, Tyler Kashuv, Shweta Lavania, Utpreksha Vaish, Ranjitha Boopathy, Ashley Middleton, Melinda Minucci Boone, Ramakrishnan Sundaram, Vikas Dudeja, Ashok Kumar Saluja

**Affiliations:** 1Departments of Surgery, Miller School of Medicine, University of Miami, Miami, FL 33136, USA; harryskcjacob@miami.edu (H.K.C.J.); shweta.lavania@med.miami.edu (S.L.); axm2962@med.miami.edu (A.M.); sramakrishnan@miami.edu (R.S.); vdudeja@uabmc.edu (V.D.); 2Sylvester Comprehensive Cancer Center, Miller School of Medicine, University of Miami, FL 33136, USA; 3School of Biosciences, Engineering and Technology (SBET), VIT Bhopal University, Madhya Pradesh 466114, India; charles.richard@vitbhopal.ac.in; 4Department of Biology, University of Miami, Miami, FL 33146, USA; rossana.signorelli@med.miami.edu; 5Department of Biochemistry and Molecular Biology, University of Miami, Miami, FL 33146, USA; txk295@miami.edu; 6Department of Surgery, University of Alabama at Birmingham, Birmingham, AL 35233, USA; utprekshavaish@uabmc.edu; 7Department of Life Sciences, Shiv Nadar University, Greater Noida 201304, India; rb130@snu.edu.in; 8Biospecimen Shared Resource, University of Miami, Miami, FL 33136, USA; mboone2@med.miami.edu

**Keywords:** pancreatic cancer, extracellular vesicles, neutrophil degranulation, granule mobilization, reteplase activity, circulating tumor cells, clusters

## Abstract

**Simple Summary:**

Circulating tumor cells (CTCs) found in the blood of pancreatic cancer patients show a worse prognosis to therapy if they are seen in clusters of cells with neutrophils or platelets or with other cell types than when they are seen as singlets. We wanted to investigate if there is a secondary mode of communication between the CTCs and neutrophils that causes them to associate. We describe for the first time an extravesicular (EV) mediated communication between CTCs and neutrophils that modulates early transcriptome changes that can cause neutrophils to partially degranulate and form associations. We also identify the protein cargo carried in such EVs and how when added to naïve neutrophils, they can modulate transcriptomic changes in neutrophils partially disarming them to form clusters rather than undergo specialized cell death, which is characterized by release of condensed chromatin (NETs) and granular contents termed as NETosis.

**Abstract:**

Tumor cells dissociate from the primary site and enter into systemic circulation (circulating tumor cells, CTCs) either alone or as tumor microemboli (clusters); the latter having an increased predisposition towards forming distal metastases than single CTCs. The formation of clusters is, in part, created by contacts between cell–cell junction proteins and/or cytokine receptor pairs with other cells such as neutrophils, platelets, fibroblasts, etc. In the present study, we provide evidence for an extravesicular (EV) mode of communication between pancreatic cancer CTCs and neutrophils. Our results suggest that the EV proteome of CTCs contain signaling proteins that can modulate degranulation and granule mobilization in neutrophils and, also, contain tissue plasminogen activator and other proteins that can regulate cluster formation. By exposing naïve neutrophils to EVs isolated from CTCs, we further show how these changes are modulated in a dynamic fashion indicating evidence for a deeper EV based remodulatory effect on companion cells in clusters.

## 1. Introduction

Pancreatic cancer is one of the most aggressive cancers, with a 5-year survival rate of less than 10% [1]. Even though surgery remains the only curative option, patients with late-stage pancreatic cancer are deemed unresectable due to distant metastases and/or extensive local vascular involvement [2]. While metastases are a main cause of cancer-related deaths, the mechanism of its spread has not been fully understood. Previously published research suggests that it is a multistep process that involves cellular detachment from the primary site, intravasation into circulation, survival in the blood stream and, finally, extravasation intro distant organs [3]. These disseminated circulating tumor cells (CTCs) travel either as single cells or as clusters. Clusters, also referred to as microemboli [4], can associate with platelets, immune cells, neutrophils, and cancer associated fibroblasts, among other cells. This association helps protect the clusters, evade immune attack and shear stress, and facilitate distant metastasis establishment [5]. Additionally, these CTC clusters display a distinct phenotype, different from single cells seen in circulation [6,7], possibly offering them enhanced survival and colony forming potential [7,8]. Moreover, these clusters seed more efficiently than single CTCs [9,10]. The presence of CTCs in circulation is mostly associated with worse clinical outcomes in several cancers such as breast [7,11], small cell lung cancer (SCLC) [10], colorectal [12], renal [13], and pancreatic cancer [14]. Thus, a deeper molecular characterization of these clusters would unlock valuable clinical data, providing a profound understanding of the molecular, immune, signaling profile of the primary tumor, which could be potentially useful for designing effective therapeutics against the specific cancer. Association of CTCs with neutrophils or cell clusters have been shown to be through cell–cell junction contacts or through cytokine-receptor pairs. Loss of function screens have identified VCAM1, CD106, and other desmosome and hemidesmosome adhesion complex genes as potential mediators of these interactions [6,7,8]. Neutrophils are the first line of defense during infections and help facilitate cancer progression [15]. Neutrophils contain several types of granules composed of several proteins, some of which have bactericidal properties, while others have important effects on innate and adaptive immune responses [16]. One of the ways neutrophils eliminate microbes is by extrusion of a mesh of chromatin fibers coated with granule derived microbial peptides and enzymes such as neutrophil elastase, cathepsin G, and myeloperoxidase. These structures are called Neutrophil Extracellular Traps (NETs), and the process is termed as NETosis. The composition of these traps are variable and it is not clear if it varies based on the stimulus [17,18]. Multiple tumor types such as breast, lung, or blood predispose circulating neutrophils to produce NETs [19]. Additionally, NETs promote CTC adhesion to capillaries and extravasation into distant organs [15]. However, not much information is available on early granulation events or neutrophil degranulation that is mediated when CTCs associate with neutrophils, allowing us to investigate other modes of communication independent of cell surface receptors. We hypothesized that the cellular mode of communication could possibly be through Extracellular vesicles (EVs) that are secreted by most cell types.

In recent years, EVs, which are membrane vesicles of endocytic origin ranging anywhere from 30–500 nm [20], have emerged as key players in intercellular communication between cells and their microenvironment. They are sturdy nanoparticles composed of proteins, cytokines, growth factors, lipids, DNAs, mRNAs, miRNAs, etc., and are released in large quantities from all cell types. Previous research from various groups on pancreatic cancer has shown that the EVs facilitate a horizontal transfer of these biomolecules [21], thereby transferring the traits from highly aggressive cancer cells to normal and indolent ones, allowing for accelerated tumorigenesis [22,23,24]. Within this context, we decided to focus on EVs that are specifically secreted from the CTCs and investigated their role as an additional mediator of cell-cell communication that would affect early changes in neutrophils and aid in the formation of CTC clusters by modulating granulatory properties of neutrophils. Our experimental methodology involved isolation of circulating tumor cells from the blood of patients with advanced pancreatic ductal adenocarcinoma and liver metastases and subjecting them to a detailed proteomic analysis to identify the components of the CTC clusters. A detailed proteomic analysis of the exosomes identified several proteins, including signature proteins that could mediate neutrophil degranulation, reteplase activity, and granule mobilization. Further, the addition of CTC derived EVs onto fresh neutrophils provide evidence for EVs to modulate the early transcriptome of neutrophils and keep it in flux over degranulation and reteplase mediated pathways. EV mediated communication in addition to surface receptor mediated recognition would help assist in CTC-neutrophil cluster formation.

## 2. Materials and Methods

### 2.1. Patient Derived Tumor Cell Lines

Three CTC lines (CM61, CF49, and HM59) were obtained from Celprogen (Celprogen, Torrance, CA, USA). Cell lines were obtained from the following patients: Caucasian male aged 61 (CM61), Caucasian female aged 49 (CF49), and Hispanic Male aged 59 (HM59). CTCs were isolated from the peripheral blood of PDAC patients with liver metastasis (mets). Equal number of cells for each patient were cultured in Medium 106 (Thermo Fisher, Waltham, MA, USA) with growth factors and 2% FBS (Thermo Fisher, Waltham, MA, USA) and then shifted to a medium with 2% exosome depleted FBS (Thermo Fisher, Waltham, MA, USA) before harvesting for EVs.

### 2.2. EV Isolation

One hundred milliliters of cell culture supernatant from all the three cell lines were collected and concentrated using a 10 kDa filter. Ten milliliters of the concentrate was incubated with magnetic EVTRAP beads (Tymora Analytical Operations, West Lafayette, IN, USA). The samples were incubated by shaking or end-over-end rotation for 60 min according to the manufacturer’s instructions. The supernatant was removed using a magnetic separator rack and the beads were washed once with PBS (Thermo Fisher, Waltham, MA, USA) and the EVs were eluted by two 10 min incubations with 100 mM of Fresh Trimethylamine (Sigma Aldrich, St. Louis, MO, USA) (TEA). Simultaneously, part of the concentrated supernatant was also subjected to Transmission Electron Microscopy (TEM) and Western blot analysis of EV markers. The nanoparticle size distribution and concentration of vesicles were analyzed on the NanoSight NS3000 (Malvern Pananalytical Inc, MA, USA) instrument using the NTA 3.1 software (Malvern Pananalytical Inc, MA, USA). The CM61 supernatant was diluted in half while the CF49 and HM59 supernatants were quantified as is.

### 2.3. Transmission Electron Microscopy (TEM)

EVs were resuspended in 2% paraformaldehyde (Electron Microscopy Sciences, Hatfield, PA, USA) and loaded on carbon Formvar-coated copper grids (Electron Microscopy Sciences, Hatfield, PA, USA), which were subsequently stained with uranyl acetate (Electron Microscopy Sciences, Hatfield, PA, USA). In the case of cells, they were fixed overnight in 2% glutaraldehyde (Electron Microscopy Sciences, Hatfield, PA, USA) in 0.1 M phosphate buffer (Electron Microscopy Sciences, Hatfield, PA, USA), post-fixed for 1 h in 2% osmium tetroxide (Electron Microscopy Sciences, Hatfield, PA, USA) in 0.1 M phosphate buffer, dehydrated through a series of graded ethanols (Pharmco-Aaper, Brookfield, CT, USA), and embedded in EM-bed (Electron Microscopy Sciences, Hatfield, PA, USA). The glass coverslip was dissolved in hydrofluoric acid (Sigma-Aldrich, St. Louis, MI, USA). Then, 100 nm sections were cut on a Leica Ultracut EM UC7 ultramicrotome (Leica Microsystems, Buffalo Grove, IL, USA) and stained with uranyl acetate and lead citrate (Electron Microscopy Sciences, Hatfield, PA, USA). The grids were viewed at 80 kV in a JEOL JEM-1400 transmission electron microscope (JEOL, Peabody, MA, USA) and images captured by an AMT BioSprint 12 (AMT Imaging Systems, Woburn, MA, USA) digital camera

### 2.4. Western Blots

The following antibodies were used to detect proteins that were present in the EV lysates; CD9 and HSP70 Cat#EXOAB-KIT-1(System Biosciences, Palo Alto, CA, USA) at a dilution of 1:200, JAK2 Cat#3230 (Cell Signaling Technology, Danvers, MA, USA) at a dilution of 1:1000, Histone H3 Cat#4499P (Cell Signaling Technology, Danvers, MA, USA) at a dilution of 1:500, and Pan-RAS Cat#3965 (Cell Signaling Technology, Danvers, MA, USA) at a dilution of 1:1000. The following antibodies were used to detect changes in neutrophils and supernatants collected post treatment with EVs from CM61 and HM59 lysates; Fibronectin Cat#2413 (Abcam, Cambridge, MA, USA) at a dilution of 1:1000, and GAPDH Cat#3683 (Cell Signaling Technology, Danvers, MA, USA) at a dilution of 1:15,000. The following antibody was used to investigate knockdown of TSG101 in the CTC cell line clones Cat#HPA006161 (Sigma Aldrich, St. Louis, MI, USA). 

### 2.5. shRNA Mediated Silencing of TSG101

shRNAs were purchased from the MISSION shRNA consortium (Sigma Aldrich, St. Louis, MI, USA). The shRNA that was most effective for TSG101 was TRCN0000380124. Lentiviral particles were made as conventionally described and transduced into the CTC lines. Cells were selected for puromycin (Thermo Fisher, Waltham, MA, USA) for 48 h and allowed to recover for a day and then imaged by TEM. Knock down was confirmed by Western blots (Sigma-Aldrich: HPA006161).

### 2.6. Antibody Arrays

CM61 cell lysates were incubated with phospho-MAPK Cat#ARY002B (R&D Systems Inc, Minneapolis, MN, USA) and phosphokinase Cat#ARY003C (R&D Systems Inc, Minneapolis, MN, USA) arrays according to manufacturer’s instructions, and Western blots were done. Representative phosphorylated proteins from both arrays common and unique were identified to show corroboration between the arrays.

### 2.7. Preparation of Samples for LC-MS

The immunoprecipitated samples were processed by Tymora Analytical Operations (West Lafayette, IN, USA). The proteins were incubated at 37 °C for 15 min to reduce the Cys residues, and alkylated by incubation in 100 mM iodoacetamide at room temperature for 45 min in the dark. The samples were diluted 3-fold with 50 mM triethylammonium bicarbonate (Sigma Aldrich, St. Louis, MI, USA) and digested with Lys-C (Wako, Ginza, Japan) at 1:100 (*w*/*w*) enzyme-to-protein ratio for 3 h at 37 °C. The samples were further diluted 3-fold with 50 mM triethylammonium bicarbonate, and trypsin was added to a final 1:50 (*w*/*w*) enzyme-to-protein ratio for overnight digestion at 37 °C. After digestion, the samples were acidified with trifluoroacetic acid (Sigma Aldrich, St. Louis, MI, USA) (TFA) to a pH < 3 and desalted using Top-Tip C18 tips (Glygen, Columbia, CA, USA) according to manufacturer’s instructions. A portion of each sample was used to determine peptide concentration using Pierce Quantitative Colorimetric Peptide Assay (Thermo Fisher, Waltham, MA, USA). The samples were dried completely in a vacuum centrifuge and stored at −80 °C. Based on the concentration, 1.4% of each peptide sample was analyzed by LC-MS.

### 2.8. LC-MS/MS Analysis

Dried peptide and phosphopeptide samples were dissolved in 4.8 μL of 0.25% formic acid with 3% (*v*/*v*) acetonitrile (Sigma Aldrich, St. Louis, MI, USA), and 4 μL of each were injected into an EasynLC 1000 (Thermo Fisher, Waltham, MA, USA). Peptides were separated on a 45-cm in-house packed column (360 μm OD × 75 μm ID) containing C18 resin (2.2 μm, 100 Å) (Michrom Bioresources, Auburn, CA, USA). The mobile phase buffer consisted of 0.1% formic acid in ultrapure water (buffer A) with an eluting buffer of 0.1% formic acid in 80% (*v*/*v*) acetonitrile (buffer B) run with a linear 60- or 90-min gradient of 6–30% buffer B at flow rate of 250 nL/min. The Easy-nLC 1000 was coupled online with a hybrid high-resolution LTQ-Orbitrap Velos Pro mass spectrometer (Thermo Fisher, Waltham, MA, USA). The mass spectrometer was operated in the data-dependent mode, in which a full-scan MS (from m/z 300 to 1500 with the resolution of 30,000 at m/z 400), followed by MS/MS of the 10 most intense ions (normalized collision energy −30%; automatic gain control (AGC)—3E4, maximum injection time—100 ms; 90 s exclusion).

### 2.9. Maxquant Label Free Quantitation

MS raw files were analyzed using the MaxQuant software (Maxquant, Martinsried, Germany) [25]. Peptides were searched against the human Uniprot FASTA database using the Andromeda search engine (Maxquant, Martinsried, Germany) [26], integrated into MaxQuant. Oxidation and N-terminal acetylation, P/T/S phosphorylations were set as variable modifications, while carbamidomethyl was fixed. Trypsin was chosen as the digestion enzyme with a maximum of 2 missed cleavages. Identified peptides had an initial precursor mass deviation of up to 6 ppm and a fragment mass deviation of 0.6 Da. The false discovery rate (FDR) for peptides (minimum of 7 amino acids) and proteins was 1%. A reverse sequence database was used in determining the FDR. For label-free protein quantification, only unique peptides were considered. A contaminant database provided by the Andromeda search engine was used. All proteins matching the reverse database or labeled as contaminants were filtered out. Label-free protein quantification (LFQ) values were obtained through MaxQuant quantitative label-free analysis [25]. The mass spectrometry data have been deposited to the ProteomeXchange consortium via the PRIDE partner repository with the dataset identifier PXD026340.

### 2.10. Neutrophil Isolation

Naïve neutrophils were isolated from human blood that was collected in EDTA tubes (Thermo Fisher, Waltham, MA, USA). 50 mL of whole blood was used for each biological replicate. The MACSexpress whole blood isolation kit (Miltenyi Biotec, Bergisch Gladbach, Germany) was used to isolate neutrophils. Purity of neutrophils were confirmed by Wright Giemsa (Thermo Fisher, Waltham, MA, USA) staining and flow cytometry. The purity was close to 99%.

### 2.11. EV-Neutrophil Co-Culture and Transcriptome Analyses

The neutrophils were separated into two groups: control and experiment. The experiment samples were incubated with freshly isolated EVs (5 μg/mL) concentrated from one of the CTC lines (CM61). A total of 10^8^ neutrophils was taken for each experimental condition. Neutrophils were collected at 10′, 30′ and 60′ after co-culture with EVs. Control neutrophils were treated with HBSS (Thermo Fisher, Waltham, MA, USA), and samples were colected after similar time points. RNA from neutrophils was isolated using the RNeasy UCP Micro kit (Qiagen, Germantown, MD, USA), quantified and processed on Clariom S (Thermo Fisher, Waltham, MA, USA) arrays using IVT Pico kit (Thermo Fisher, Waltham, MA, USA). Extracted RNA underwent the GeneChip 3′ IVT Pico workflow prior to hybridization, washing, staining, and scanning of the Clariom S array. The 3′ IVT Pico workflow is completely described in the kit user guide, but is a multi-step process where extracted RNA are prepared to be hybridization-ready cDNA targets. Extracted RNA samples undergo the stepwise process of first-strand cDNA synthesis using reverse transcriptase of the RNA, cleanup of excess primers, 3′ adaptor synthesis, pre-IVT amplification, cRNA amplification, cRNA purification, 2nd-cycle ds-cDNA synthesis, template RNA removal, ds-cDNA purification and quantification, fragmentation and terminal Biotin labeling, and finally hybridization to the array. Following hybridization, the Clariom S arrays are washed, stained, and scanned using a GeneChip Scanner 3000 (Thermo Fisher, Waltham, MA, USA). The microarrays were processed at the Microarray Research Services Laboratory (MRSL) at Thermo Fisher Scientific, (Thermo Fisher, Santa Clara, CA, USA). Raw data from the Human Clariom S arrays were analyzed on the Transcriptome Analysis Console (Thermo Fisher, Waltham, MA, USA). The quality of the experiment was assessed based on the values of pos vs. neg auc and pm mean, which were calculated using the Affymetrix^®^ Expression Console software (Thermo Fisher, Waltham, MA, USA). CEL files were processed for each replicate and experimental condition using the Affymetrix^®^ Transcriptome Analysis Console software 4.0 (Thermo Fisher, Waltham, MA, USA). The CEL files were then subjected to normalization using the Signal Space Transformation-Robust Multiarray Analysis (SST-RMA) method [27] to generate CHP files. To analyze significantly differentially expressed gene (fold change ≥ ± 2, *p* value < 0.05). There was no significant difference between the neutrophils collected over different time points, so all comparisons were done against the 2-min time point of neutrophils treated with PBS. Neutrophil sample collected at time point 2-min is represented as 2′, while neutrophils treated with EVs and collected at time points 10′, 30′, and 60′ are, respectively, represented as EV10′, EV30′, and EV60′.

### 2.12. Neutrophil Granulation and NETosis Analysis

Naïve neutrophils were isolated using the MACSexpress (Miltenyi Biotec, Bergisch Gladbach, Germany) whole blood isolation kits. Purity of neutrophils was confirmed by Wright Giemsa staining (Thermo Fisher, Waltham, MA, USA) and flow cytometry. The purity was close to 99%. Overall, 10^9^ neutrophils were treated with CM61 EVs or HM59 EVs for 2′, 10′ 30′ and 60′ in HBSS buffer. Supernatants and neutrophils were collected post treatment along with untreated controls and controls that were incubated in HBSS for 60′. Then, 1000 ug of cell supernatants from the 2′, 10′, 30′, and 60′ neutrophils treated with CM61 EVs were subjected to the chemokine analysis using the Human Chemokine array (R&D Systems, Minneapolis, MI, USA). Arrays were quantified using ImageJ software (NIH, DC, USA).

### 2.13. Real-Time PCR (qPCR) and Data Analyses

Quantitative Real time PCR was performed using the standard SYBR Green method using a Lightcycler 480 (Roche Applied Science, Penzberg, Germany) employing standard PCR conditions. All the reactions were performed in triplicates with control neutrophils without treatment and neutrophils treated with EVs for time durations of 2′, 10′, 30′, and 60′. The expressions of the target genes were normalized to endogenous control 18S rRNA. For data analysis, the result files were extracted in XLSX format, and cycle threshold (Ct) values were averaged out. The mean Ct values of target genes were normalized to the mean Ct values of endogenous control for corresponding samples, which gives ΔCt value. ΔΔCt was calculated as (Test ΔCt—Control ΔCt), and the fold change was calculated using the formula 2^(−ΔΔCt)^, such that the reference sample would have a fold change value of 1. This way, the differential expression for a specific gene is calculated as compared to control. GraphPad Prism 5.0 software (GraphPad Software, San Diego, CA, USA) was used for plotting the data for expression of each gene.

### 2.14. Enrichment Analyses

All enrichment analyses with the gene lists were done on ShinyGO [28], an intuitive graphical web application to visualize pathways from different databases. Only pathways that had a *p*-value of ≤ 0.5 were considered. The Reactome [29] and NetPath [30] databases were primarily used for all analyses.

## 3. Results

### 3.1. Circulating Tumor Cells Secrete Extracellular Vesicles

Despite recent strides in understanding the nature and role of CTCs in promoting cancer, not much is known about how the circulating tumor cells communicate with other cells in circulation. Is it a process that is solely based on surface protein interactions [6,7] or soluble mediators, or can the CTCs communicate by other modes as well? Thus, we decided to investigate the possibility of CTCs being able to secrete EVs and whether these EVs could be the effectors in this mechanism. Pancreatic cells are known to secrete chemokines and cytokines that modulate several immune reactions or assist in the recruitment of several cell types by the tumor [31,32,33,34]. Recent studies have identified exosomes or extracellular vesicles as efficient mediators of this cell–cell communication [23,35,36].

However, there are no reports of CTCs secreting EVs primarily due to their low abundance or short life spans in circulation. Thus, in the present study, in order to identify whether the CTCs secreted EVs, we investigated cell lines isolated from PDAC patients with liver metastases and cultured them for 4–5 generations in low attachment plates and low serum containing medium. These cell lines were positive for CD133, CD44, and cytokeratin markers. We observed a mixed population of cells growing in culture, showing two distinct morphologies when plated on adherent plates. Single cell sorting of cells could not separate the two cell types with different morphologies with one cell type giving rise to the other. All three cell lines exhibited the same morphologies. For all experiments, EVs were collected from a mixed pool of cells from individual patients with differing morphologies. Cell culture supernatant was collected from cells that were grown in exosome-depleted FBS for 48 h, cleared of large debris, and concentrated using a 10 kDa filter. We identified several EVs of differing sizes though the yield of EVs was about 10% in quantity when compared to established pancreatic cancer cell lines such as Panc02. Additionally, transmission electron microscopy identified EVs of various sizes present in the concentrated supernatant (Figure 1A). Nanosight tracking analysis of supernatants from cell lines showed a slight variation in the actual particles/mL from the three different cell lines (Figure 1B). The CM61 cell line had 8.62 × 10^8^ +/− 3 × 10^7^ particles/mL with different size ranges up to 500 nm, the CF49 cell line with a concentration of 4.88 × 10^7^ +/− 5.6 × 10^6^ particles/mL ranging up to 400 nm, and the HM59 cell line with a particle/mL count of 3.99 × 10^8^ +/− 1.85 × 10^7^ ranging up to 400 nm. Relative light scattering intensities is plotted against particle size in Figure 1B. A detailed NTA analysis is given in Appendix A. These findings are the first reports of CTCs secreting EVs in an in vitro system.

In order to establish whether a signaling component was also present in the EV population, we analyzed the expression of pancreatic cancer specific markers in addition to conventional EV markers. Western blot analyses were used to characterize EV markers such as CD9, HSP70, and Histone H3 in addition to PDAC signaling markers such as Jak2 and Pan-RAS in EV preparations (Figure 1C). Detailed quantification for all western blots are provided in Appendix A.

### 3.2. Secretion of EVs from the CTCs Is an Active Process

Exosomes have been classically defined as secreted EVs corresponding to intralumenal vesicles (ILVs) of multivesicular bodies (MVBs) formed during maturation of endosomes. Some are destined for degradation, while others can fuse to the plasma membrane, allowing for the secretion of the ILVs as exosomes [37]. In order to eliminate the presence of cellular artifacts or cellular fragments released by the apoptotic CTCs, it was necessary to modulate the EV secretion pathway and identify if the phenotype of the cell changed. To this end, we decided to silence the expression of *TSG101* (Tumor Susceptibility Gene 101), an essential component of the Endosomal Sorting Complexes Required for Transport-I (ESCRT-I) complex using shRNAs. TSG101 is an essential protein in the ESCRT secretion pathway. It has a major role in endosome to cytosol release of the MVBs by presumably controlling the back-fusion process in early viral studies [38]. Due to the mixed population of cells, we single cell sorted the knockdown cells and expanded them and probed for the expression of TSG101. We identified a single clone with higher knockdown of endogenous TSG101 (Appendix A) that was used for imaging studies. Our results show EV release from the membrane is inhibited in *TSG101* knocked down cells (KD) with respect to control cells transduced with lentiviruses expressing a scramble shRNA. The KD CM61 cells show a very distinct convoluted membrane structure that contained several invaginated bodies on closer investigation with TEM. TSG101 is a key component involved in sorting ubiquitinated receptors into MVBs and is involved in the membrane invagination process that results in the formation of internal vesicles [39,40,41]. TEM was done to determine changes in plasma membrane with respect to inhibition of the Endosomal Sorting Complexes required for Transport (ESCRT) pathway. We noted that in CM61 CTC transduced with scramble shRNA a Figure 1E(i), the plasma membrane did not have any invaginations or vesicles attached to the membrane and showed a smooth structure characteristic of a normal cell. However, on knocking down *TSG101*, the CM61 cell lines, as shown in Figure 1E(ii), exhibit a distinctly serrated membrane, indicating that the EV secretion in the CTC has been affected with EVs not being released into the supernatant and accumulating on the surface of the plasma membrane. Thus, our results provide evidence for an active secretion pathway involved in CTCs. Inhibiting *TSG101* makes cells accumulate vesicles on the surface and in turn senesce over time in culture.

### 3.3. CTC EVome Contains Both a Protein and Phosphoprotein Cargo

Mass spectrometric characterization of EVs identified a total of 825 proteins across all the three cell lines. Among the 825 proteins identified, 787 were identified in the CM61 CTC line, 796 in the CF49 CTC line, and 801 from the HM59 CTC line. Additionally, we found 765 proteins that are common among the three cell lines Figure 1D. In order to understand the function and location of these proteins, a Gene Ontology (GO) analysis was done characterizing the proteins into different groups according to their distribution in the cellular components, biological processes, and molecular functions. Appendix A lists the 20 most significant annotations identified for each of these functions. Looking into the biological processes, the proteins were mainly enriched in the protein component biogenesis and assembly, proteins containing complex subunit organization, and mRNA metabolic processes. These findings correlate with the results from an earlier study in a PDAC cell line BxPC3 [42] indicating that a common phenotype possibly exists among the pancreatic cancer cell lines and the CTCs. In the enrichment of cellular components, the Differentially Expressed Genes (DEGs) were mainly associated with extracellular vesicle, extracellular organelle, or exosome cellular components with significant enrichment of proteins observed in the extracellular space. This observation clearly adds to the evidence that we were sampling a pool of vesicle-related proteins. GO enrichment analysis of molecular functions for these 825 proteins enriched for protein involved in RNA binding, cell adhesion molecule binding, cadherin binding, and cytoskeletal protein binding. It is important to note that we have identified proteins that carry a unique signature for cellular recognition. A complete list of proteins identified from all cell lines is provided in the Appendix A and GO enrichment reports are listed in Appendix A.

KEGG (Kyoto Encyclopedia of Genes and Genomes) pathway enrichment of the proteins is shown in Appendix A. Some of the top pathways identified in our screen are associated with focal adhesion, RNA transport, tight junction related pathways, spliceosome, ECM-receptor interaction, and PI3K-AKT signaling pathways, etc. Interestingly, multiple groups have reported a strong correlation between the PI3K pathway, which is essential for various cellular functions (such as endothelial cell sprouting for angiogenesis, T cell differentiation and homeostasis, fibroblast associated resistance, and macrophage transcriptional reprogramming [43,44,45,46]) and pancreatic cancer. Additionally, several signaling pathways converge at the MAPK and PI3K signaling pathway in pancreatic cancer which are effectors of multiple cellular response in the cell. As reported in earlier studies, the active intracellular signaling cues characteristic of these signaling pathways are released into the extracellular milieu.

In order to investigate whether the EVs show significant signs of activated signaling in conserved and preserved sequences, we performed a phosphoproteomic characterization of the EVs isolated from the three cell lines. We identified 660 proteins among the three cell lines that contain pS/T/Y modifications (Figure 1D). Additionally, a total of 589 proteins had phosphorylations in the CM61 cell line, 595 proteins in the CF49 CTC line, and 601 in the HM59 cell line EV populations. As there were no significant differences between the three cell lines, we looked much closer at the proteins that had P/T/Y modifications identified in all the three cell lines. It was interesting to note that there were nearly 560 modified proteins that were common to all the EVs isolated from the three cell lines. We also performed a NetPath pathway analysis to determine additional pathways that could be significantly enriched as well. Our results identified multiple pathways associated with reteplase action, Inositol metabolism, Trehalose degradation, and Ardeparin action that were significantly enriched. Tissue plasminogen activator (PLAT) is one of the constituents of the EV proteome. Reteplase is a modified non-glycosylated form of tissue plasminogen activator (tPA) that catalyzes the conversion of inactive plasminogen proenxyme into the active plasmin protease by cleavage of a Arg-Val bond; this helps in degrading fibrin matrices [47,48]. For sake of simplicity, we mention all proteins that are enriched for inducing breakdown of fibrin matrices or clots such as PLAT and other proteins of the pathway as “Reteplase activity” modulating proteins. Overall, our data indicate for the first time that EVs carry a signature for modulating reteplase activity that could explain the changes in the fibrin-based protective cloak that has been shown in earlier studies in helping CTCs evade immune detection [49,50,51]. A complete list of phosphoproteins identified from the cell lines is given in Appendix A.

### 3.4. Active Signaling Signature Is Conserved in the EV Cargo

Our next question was to investigate whether the EV cargo contains certain conserved residues between the EVs and the cells. To accomplish this, we prepared lysates of the CM61 cell line, and analyzed the phosphorylation pattern using phospho-MAPK and phosphokinase arrays (Appendix A). We observed that the cell line had an active MAPK pathway and the EVs isolated from the cell line contained similar phosphorylation patterns with respect to AKT1 and ERK2. The phospho-antibody arrays detected pERK1 (T202/Y204) and pERK2 (T185/Y187) and pAKT (S473/T308). For higher sensitivity, we performed a mass spectrometry analysis which identifying additional activated signaling signatures of the MAPK pathway in the EVs. We identified the same sites for AKT1 (T308) and ERK2 (T185/Y187) in the proteome of the EVs. A summary of the phospho-MAPK array, the identified sites, and their biological significance is shown in Table 1. Taken together, we believe that the EVs that are secreted from CTCs into the blood stream could also indicate potential signaling patterns representative of the primary tumor or in the cells in circulation, providing thus a snapshot of signaling patterns at the primary site. Additional phospho sites identified on all proteins are shown in Appendix A.

### 3.5. Neutrophil Degranulation and Reteplase Activity Pathways Are Modulated on Exposure to CTC EVs in Neutrophils

A Reactome analysis of the 765 proteins common to all the three different CTC lines was undertaken in order to identify the signaling pathways that could be regulated by these proteins, thus providing a better understanding of the cluster formation between CTCs, blood, and immune cells. Our analysis shows that there were 93 and 37 proteins that were enriched in the neutrophil and platelet degranulation pathways, respectively. A complete list of pathways enriched along with their genes is provided in Appendix A. The interplay between the CTCs and platelets has been known to play a very important role in metastases by enhancing tumor cell survival, tumor-vascular interactions, and escape from immune surveillance [52]. Earlier studies have identified populations of CTCs extensively covered by platelets [52] as well as CTC-neutrophil [6,53] clusters. As these clusters evade conventional positive and negative enrichment mechanisms, further characterization of these cells and the mechanism of formation of these clusters needs to be understood in order to better decipher bloodborne metastasis. Moreover, on a NetPath analysis, we observed that 62 of the identified proteins were part of the reteplase activity pathway. Thus, taken together, our data suggests the possibility that the EV proteins could modulate neutrophil behavior directly.

The discovery of proteins that contribute to degranulation and reteplase activity led us to investigate temporal changes to the degranulation pathways in neutrophils. To this end, freshly isolated naive neutrophils from normal donors were subjected to different experimental conditions, the controls were treated with HBSS and samples were collected at different time points and the experimental conditions were treated with EVs from the CTC line CM61. Untreated neutrophils were designated as the baseline and compared to neutrophils that were treated with EVs. The control neutrophils, incubated with HBSS and collected at different time points, were similar in their expression profile with no significant differences (more than 96% of all genes identified showed no changes). Hence, for ease of analysis, all comparisons were with the 2′ time point of control neutrophils. To assess the similarities and differences in the transcriptomic profile of these different time points and control samples, PCA plot analyses were performed on the samples. Except for the 10′ exosome treated time points pre normalization, the rest of the samples clustered close to each other (Figure 2A).

A comparative analysis of the transcriptome of the neutrophils treated with EVs collected at 10′ vs. neutrophils at 2′ showed 1676 genes (57.59%) that were upregulated, while 1234 (42.41%) genes were down regulated. Of the upregulated genes, 62.53% were multiple complex genes (genes whose classification locus could not be defined), 24.82% coding, 8.89% noncoding, 3.34% unassigned, and 0.42% were pseudogenes. Among the 1234 downregulated genes, 88.01% of the genes were multiple complex genes, 8.35% were coding genes, 3.08% were non-coding, and 0.57% genes with unassigned functions. In comparison with the 30′ time point, there were only 638 genes (86.36% multiple complex, 8.46% coding, 4.08% noncoding, and 1.1% genes with unassigned functions) that were upregulated and 511 genes that were downregulated (78.28% multiple complex, 8.41% coding, 8.02% noncoding, and 5.28% genes with unassigned functions). Analyzing the final time point, where the control neutrophils were compared with the co-cultured neutrophils at 60′, we identified 428 genes (86.45% multiple complex, 9.58% coding, 3.27% noncoding, and 0.7% genes with unassigned functions) upregulated and 462 genes downregulated (77.49% multiple complex, 8.87% coding, 7.58% noncoding, and 6.06% genes with unassigned functions). A detailed summary of the gene percentages is shown in Table 2. Scatter plots of the three comparisons are shown in Figure 2B. There is an observable decrease in the genes that are differentially regulated from the 10′ to the 60′ time point.

This could be indicative of the neutrophils returning to the baseline transcriptomic signature post activation by the EVs. Hierarchical clustering of all the samples (Figure 3A) indicates the similarity between the biological duplicates for each experimental condition with the EV from the 10′ time point clustering very differently with respect to the other EV sample time points at 30′ and 60′. This indicates that there are major transcriptomic changes that are initiated within the first 10 min with the neutrophils trying to get back to the normal gene stasis in later time points. Expression values for the genes identified in each experimental subtype is presented in Appendix A.

In order to understand whether these differentially modulated genes modulate certain specific signaling networks, a pathway analysis using curated pathways from the Reactome and NetPath was performed. In comparison to the naïve neutrophils, within the first 10 min of exposure to EVs, there is an increase in the olfactory, G alpha, and the GPC receptor signaling followed by keratinization, defensins, collagen formation, and trimerization pathways, along with pathways that are responsible for extracellular matrix organization. Interestingly, the pathways responsible for platelet degranulation are downregulated. Nearly 98 genes that characterize this pathway are suppressed. Furthermore, the innate immune system pathways are also down regulated. It is interesting to note that the translation initiation, elongation, and termination are suppressed, and so is RNA metabolism. With the next 30 min of interaction with EVs, there is a period of recovery where protein initiation, elongation, and termination pathways are upregulated again. The neutrophils go through a period of activated RAF-MAPK signaling and increase in transcription of genes responsible for repairing and restoring the balance in these cells. Based on these findings, we propose that the platelet degranulation pathway genes (21 genes) are upregulated primarily as a response to restore balance to the earlier suppression on EV exposure. However, the balance does not tilt in the favor of the neutralization of this pathway as the activation is only partial and there are more than 25 genes still being activated in the pathway tilting the balance towards suppression of the pathway in general. At 30 min of exposure to EVs, there is also a distinct suppression of the vesicle-mediated transport and the immune-related genes. At 60 min of exposure, NOTCH, JNK, and TLR cascades (TLR2, TLR4, TLR5, TLR7, TLR8 and TLR10) are activated as well. There is continued expression of genes modulating the protein expression being activated. Interestingly, the interplay between the neutrophil degranulation pathways continues with more genes being downregulated (23 genes) than expressed (17 genes) at 60′ pushing and maintaining the neutrophils to suppress the degranulation pathways. As for the reteplase activity pathway, within the first 10 min nearly 56 genes that modulate this pathway are downregulated, while at 30 min, there are 23 of these genes whose expression is upregulated to return the transcriptome to stasis. No other significant change was observed other than suppression of several biochemical pathways from the curated sets of genes from NetPath. A complete list of the significant NetPath and Reactome pathways annotated from the overexpressed or suppressed gene sets is given in Appendix A with the expression values of genes of the reteplase activity and neutrophil degranulation pathway identified from this study shown in Appendix A. Expression heat maps of the genes identified in this study for reteplase and neutrophil degranulation specifically are shown in Figure 3B,C. We also wanted to see if the neutrophil degranulation genes are specifically identified in one particular component of the granulation pathway or not. A Reactome analysis indicates distribution across all granule specificities being downregulated at time point 10′ with lesser activation across the granular types in preceding time points. A detailed list of genes and enrichment in the precise granular component is shown in Figure 4.

### 3.6. CTC EVs Modulate Granule Mobilization on Mature Human Neutrophils

Mature neutrophils are recognized by a segmented nucleus and an abundant number of cytoplasmic granules that contain a variety of proteins that play vital roles in engulfment, killing of foreign particles, help in cell–cell interactions, signaling, and modulation of the surrounding environment and extravasation [54]. Cytoplasmic granular fusion leads to compound exocytosis in two ways, either through the intra granular fusion in the cytoplasm or through the sequential fusion of granules to the same site on the plasma membrane. It is a tightly regulated process that is not completely understood. Several proteins regulate the formation of these cytoplasmic granules with the SNARE and RAB proteins playing a crucial central role. There are three types of cytoplasmic granules that are released by neutrophils which are the primary or the azurophil granules, secondary or specific granules and tertiary or gelatinase granules [55]. The microbicidal compartment being the primary granules, while secondary and tertiary granular proteins degrade the ECM and tertiary granules modulate neutrophil biology such as diapedesis or extravasation, cell adhesion, and superoxide anion production [56,57,58,59,60]. Markers were selected for analysis for each specific granule type; azurophil granules (CD63), specific granules (CD15 and CD66b), and secretory vesicles (CD13, CD14, and CD18) [61]. The expression of CD63 in neutrophils treated with EVs for 2′ and 10′ minute time points compared to control is less than 0.20-fold, which increases to 0.62-fold at 60′ post EV treatment [Figure 5(i)]. Likewise, expression of CD66b [Figure 5(ii)] is around 0.40-fold at the 2′, 10′, and 30′ time points and increases to nearly 0.88-fold at 60′. The secretory vesicle granule proteins CD14, CD13, and CD18 [Figure 5(iii), (iv), and (v)] show similar trends in changes in fold change; (CD14:0.40-fold at 2′, and 10′ treatments and becomes equal to expression of control at 30′ and then decreases to 0.58-fold as compared to control at 60′; CD13 and CD18: The expression fluctuates between 0.72-fold and 0.50-fold, respectively, at 2′ treatment, decreasing in both cases at 10 min and increasing again at 30′ and falling below at 60′ of treatment).

EV treatment of neutrophils reduces the expression of granular markers and then subsequent time points show that there is gradual recovery of expression levels. These observations are in line with the working hypothesis suggested by Mollinedo [54] in cancer metastasis involving “neutrophil-drawn carriages” that are fueled by granule mobilization. These cancer cells, thus, evade the immune system, and the differential granule exocytosis could direct the “neutrophil-drawn carriages” to the distant organ. Based on our observations, we hypothesized that the EV-mediated communication between the CTC and the neutrophil could modulate granularity profiles or the secretome in the cluster and this well-orchestrated ballet of granule mobilization, reteplase activity regulation and suppression of neutrophil degranulation could determine the cluster targeting to metastatic site and extravasation into the distal site. We examined the levels of secreted fibronectin as a marker of NETosis as mentioned in an earlier study [62] to quantify the extent of degranulation in neutrophils on exposure to EVs from CTCs. We observed contradictory results with respect to secreted fibronectin with the levels being very variable among the two cell lines (Appendix A). In the chemokine analysis, we observed an increase in CXCL7, CXCL4, and CXCL12, while all other chemokines did not change on extended treatment with CTC EVs (Appendix A). CXCL7 is a platelet specific chemokine [63,64] that is released from platelet α granules under platelet degranulation, however, we could find no instance of CXCL7 being released from neutrophils in the literature. It helps in neutrophil recruitment to the site of inflammation [65]. On the other hand, CXCL4, though not a prominent leukocyte chemoattractant [66,67], has been shown to induce neutrophil secondary granule exocytosis and release of matrix degrading enzymes that could help in neutrophil penetration of infected or injured tissue [68]. CXCL12, on the other hand, is also part of the CXCL12–CXCL4 pathway that modulated neutrophil activity during acute lung injury [69]. While more work still needs to be done to verify these results, it is an observation of ours with respect to neutrophil protein secretion on exposure to CTC EVs.

## 4. Discussion

Pancreatic cancer is one of the deadliest cancers, primarily due to the lack of defining early symptoms, which are often missed during the early screening tests, and serious symptoms manifest only when the disease has progressed to an advanced stage. Pancreatic cancer metastasizes to several organs such as the liver, stomach, diaphragm, and other sites with rich vasculature, making surgery a difficult option for treatment at later stages. Circulating tumor cells are a source of metastases, but only a small fraction of them eventually succeed in reaching the distant metastatic site [70]. Over the past decade, many studies have identified CTC clusters in different cancer types and increasingly it has become evident that these CTC clusters are significantly more metastatic than single CTCs and are always associated with a worse prognosis. Therefore, there is a greater need to investigate CTC crosstalk with others cells that form these clusters to further understand cancer progression and metastasis. Recent studies also indicate that these clusters exhibit very distinct features with respect to their phenotype, gene expression signature, and mode of dissemination. Additionally, several studies have shown the need for cell–cell contact between the CTC and the associating cell to ensure cluster formation. Clusters are generally composed of several cell types such as blood cells, immune cells, fibroblasts, etc., drawing the need for a more universal mode of communication between cells types in addition to cell–cell junction interactions. To this effect, we have investigated whether EVs can act as facilitators of the CTC clusters formation by modulating the gene expression in the neutrophils associated with clusters. The CTC EVome revealed protein signatures that can modulate cellular communication with neutrophils and platelets. Additionally, treatment of naïve neutrophils with fresh EV extracts from the CTCs showed distinct remodulatory effects such as granule mobilization and degranulation influencing the fate of the neutrophils by suppressing NETosis.

The mechanism of how CTC clusters with neutrophils has been a matter of study for several years with several studies indicating physical contact with receptors or proteins as being the primary mode. Association of neutrophils with CTCs would induce NETosis as reported in tumor spheroid models in colon carcinoma, wherein formation of neutrophil extracellular traps protect clusters from T cell or natural killer cell mediated toxicity [71]. In our experience with PDAC and colon CTCs collected from peripheral blood of patients, we do not observe any visible NETs and observe mostly intact CTCs or clusters indicative of NETosis not being a primary mode for the formation of clusters. However, maybe a “mild” form of NETosis that is reversible could occur but not have any visible characteristics of extended DNA structures. The proteomic content of CTC EVs show enrichment of proteins that could induce platelet and neutrophil degranulation, providing further evidence for maybe a milder degranulatory process that could be involved or also adding to the hypothesis a reversible phenomenon. This could explain in effect as partial degranulation to associate with CTCs as soon as they are released from the primary site into circulation, followed by reversal of the granularity on reaching the target site or primed pre metastatic niche. This partial degranulation has been recently termed as “neutrophil disarming” [72], indicating that the neutrophils can be disarmed and not form NETs with changes in their proteome and granules. Early mRNA transcriptomics of naïve neutrophils treated with EVs show that degranulation and reteplase activity pathways are differentially regulated. We have shown transcriptomic evidence that the EVs do not allow neutrophils to completely degranulate and form NETs, as there is a dynamic change in the expression levels of proteins that constitute the degranulatory/reteplase pathways in neutrophils. Most importantly, we provide transcriptomic evidence for the first time that the EVs secreted from the CTCs can modulate “granule mobilization” that possibly allows for the hypothesized formation of “neutrophil-drawn carriages” [54], thereby mediating cluster targeting to the metastatic site eventually.

We do realize that there are limitations to this study and that are considerations we are exploring in future studies; the experimental approach in this study limit the work that can be done in this regard allowing us to only focus on a two-cell cluster type in isolation with emphasis given to unidirectional EV mediated communication between the CTCs and neutrophils and not the other way round, given neutrophils are also known to secrete exosomes [73]. We investigate only a unidirectional transfer of EVs from the CTCs to neutrophils and not vice versa. We postulate that EV mediated communication is a novel mode of communication between CTCs and neutrophils in addition to surface receptor mediated modes of signal transfer. CTCs are cells that are difficult to culture in vitro, and there are changes associated with the cells on exposure to plastic or cell culture media. We compared the EVome from CTCs (Bonafide accessions could be assigned only to 759 of the 765 proteins common to cell lines for comparison across different datasets) with published data from Capan-2 PDAC cell line [74] and the surface proteome of Exosomes from multiple cell lines [75]. More than half of the proteins were similar to those identified from the different cell lines (Appendix A). Since there is no precedent for us to compare with CTC EVome in circulation, we compared to total serum EVome [76] only to identify a meagre 11% of proteins (Appendix A). Unfortunately, we could not experimentally verify if the EV knock down cells could form less clusters in PDX models given that TSG101 knock down caused cells to stop proliferating, indicating that EV secretion was vital. Knock down or knock out of other EV pathway modulating genes was not as effective in eliminating the EV secretory pathway, and we could not generate a complete EV null cell type. We also tried previously published EV inhibitors, but did not see an effect in EV secretion from the CTC cell lines. However, on serum starvation and or treatment with Erlotinib to investigate if the CTC EV phosphoproteome changes based on the basal state of the CTCs, we observed a change in the distribution and sizes of the EVs through NTA analysis (Appendix A). We could not quantify the changes on a phospho-MAPK array as it was below the detection limit. This would also add to the superiority of using a mass spectrometric approach to investigate EV cargo as the gold standard in research or in a diagnostic setup.

However, taken together, our experiments provide evidence that the CTC EVs induce changes in neutrophils within the first few minutes of contact. This is in addition to interaction of neutrophils and other cells that is mediated by cell surface receptors (VCAM1, CD106, and other desmosome and hemidesmosome adhesion complex genes). We hypothesize that this two-way communication allows for neutrophils to move from a partially “sticky” disarmed state on association to perhaps dynamic changes in receptors or cell–cell contacts that decrease “stickiness”, allowing cells to dissociate on reaching their target site. In a disease such as pancreatic cancer, where metastases form the major cause of death, a dual approach can be used to target these clusters. Firstly, targeting CTC surface receptors involved in association with neutrophils and or other cells and secondly targeting EV secretion in CTCs or neutrophils; the latter being more challenging due to the universal nature of EV secretion from cells and cancers.

## 5. Conclusions

In summary, our studies demonstrate that extracellular vesicles secreted from pancreatic CTCs induce changes in the transcriptome of neutrophils in clusters. Analyses of CTC derived EV cargo showed enrichment of proteins that are associated with degranulation of neutrophils and platelets. This is a dynamic phenomenon and correlates strongly with PLAT (Reteplase) activity. These data suggest that EVs secreted by the pancreatic cancer CTC can remodel the tumor microenvironment by influencing neutrophil activation. Such functional changes may allow for the formation and break up of neutrophil-CTC clusters and potentially facilitate distant metastasis.

## Figures and Tables

**Figure 1 cancers-13-02727-f001:**
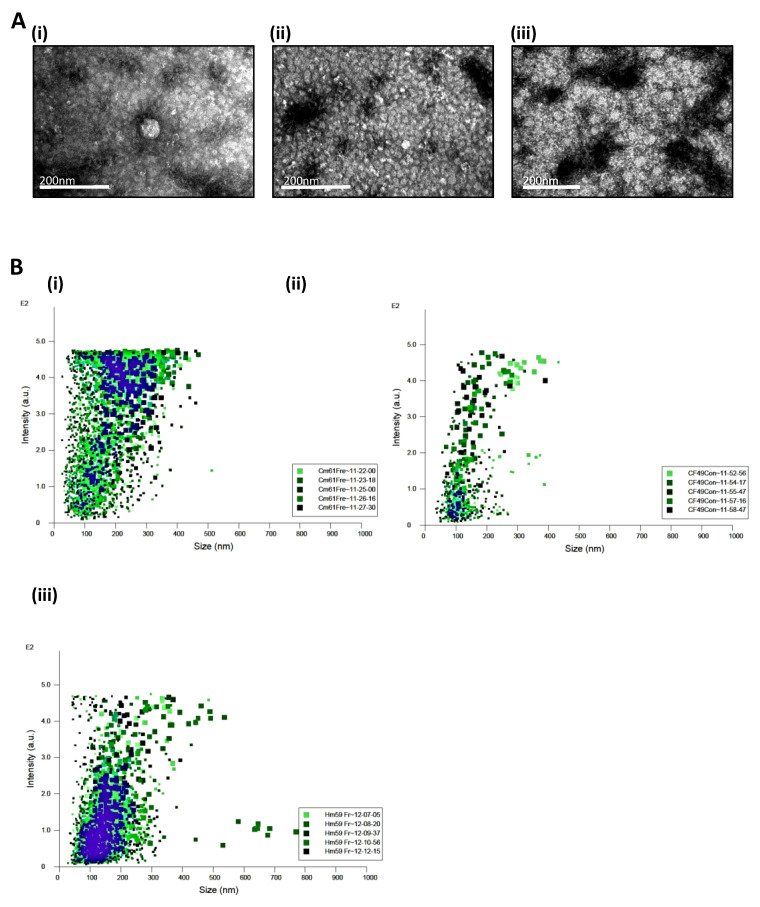
Characterization of extracellular vesicles secreted from patient derived CTC lines; (**A**): Representative images from transmission electron microscopy of concentrated EVs secreted from (**i**) CM61, (**ii**) CF49, and (**iii**) HM59 cell lines (Magnification ranges from 250,000× to 300,000×); (**B**): NanoSight (Malvern Pananalytical Inc, MA, USA) analysis of size and concentration of EVs secreted from the CTC lines with size in nm on X axis versus scattering intensity on Y axis for (**i**) CM61 CTC cell line, (**ii**) CF49 Cell line, and (**iii**) HM59 cell line. (**C**): Western blot analysis illustrating the expression of characteristic EV markers CD9, HSP70 (System Biosciences, Palo Alto, USA) and Histone H3 (Cell Signaling Technology, Danvers, USA), and PDAC signaling markers such as JAK2 (Cell Signaling Technology, Danvers, USA) and Pan RAS (Cell Signaling Technology, Danvers, USA). Lane 1: Lysate of EVs from CM61; Lane 2: Lysate of EVs from CF49; Lane 3: Lysate of EVs from HM59; (**D**): Distribution of (**i**) proteins and (**ii**) phosphoproteins identified in the three PDAC patient derived CTC cell lines CM61, CF49, and HM59; (**E**): Knockdown of TSG101 ESCRT0 protein in CM61 cell line was done with shRNAs from the TRC consortium. Silencing of this essential ESCRT pathway protein does not allow pinching of EVs from the surface of membrane, allowing them to accumulate on the membrane surface: (**i**) TEM of cell surface of CM61 cell line transduced with scramble shRNA. Inset image is magnified image of the CM61 plasma membrane and the arrows indicate a smooth plasma membrane with no accumulation of budding bodies or vesicles. (**ii**) CM61 cell line transduced with shRNAs targeting *TSG101*. Inset image shows a smooth plasma membrane being absent with accumulation of vesicular bodies and no secretion due to a defect in the secretion pathway. The arrows indicate the vesicles that are attached to the plasma membrane and are not released into the supernatant indicative of a defective secretion pathway.

**Figure 2 cancers-13-02727-f002:**
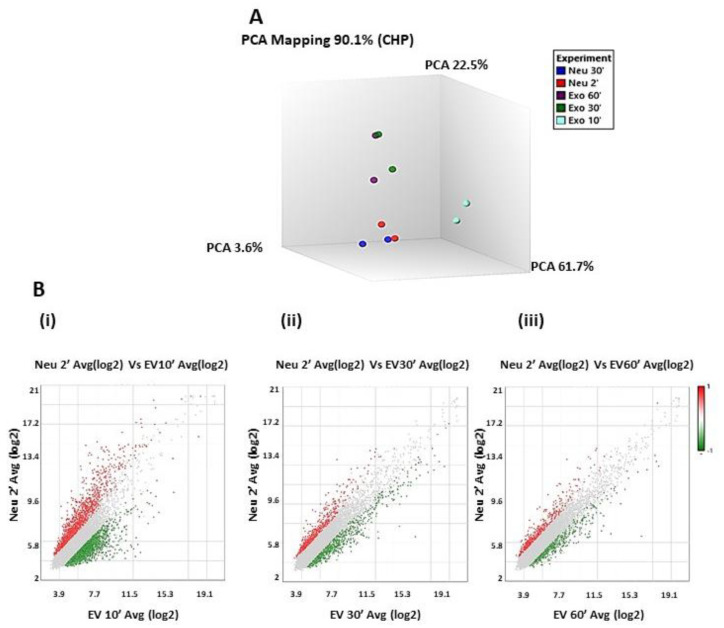
Transcriptomic analysis of CTC EVs incubated with neutrophils; (**A**): Principal Component Analysis (PCA) of microarray data obtained by hybridization of mRNA extracted from control neutrophils at time points (2′ and 30′) and experimental samples, i.e., neutrophils treated with EVs and collected at time points 2′, 30′, and 60′; (**B**): Scatter plot representing the average (log2) fold change of genes expressed in neutrophils at 2 min vs. average (log2) fold change of genes expressed in neutrophils treated with EVs from CM61 cell line at time points (i). 10′, (ii). 30′, and (iii). 60′.

**Figure 3 cancers-13-02727-f003:**
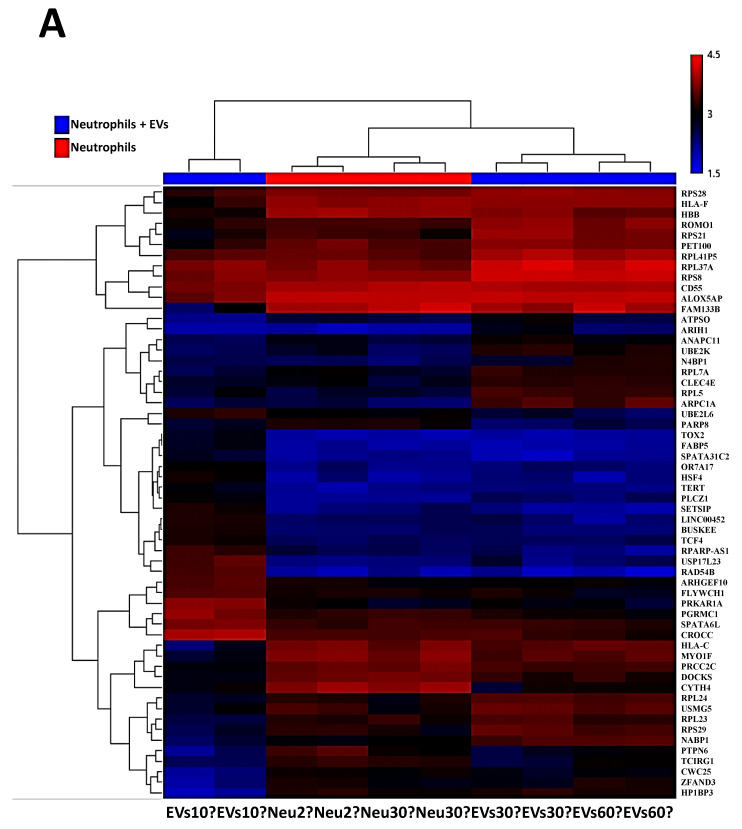
(**A**): Hierarchical clustering heatmap representing differential gene expression between neutrophils at 2′ and 30′ and neutrophils treated with EVs from CM 61 cell line. The sample represented as “Neu/EVs” denotes a biological duplicate. The Log2 fold changes are represented here ranging from a maximum of 4.5 fold to a low of 1.5 fold. The top differentially expressed genes are shown on the right; heatmap of gene expression of constituent genes in (**B**) reteplase and (**C**) neutrophil degranulation pathways with basal levels shown as Neu 2′ and neutrophils treated with EVs at different time points (EVs 10′, EVs 30′, and EVs 60′).

**Figure 4 cancers-13-02727-f004:**
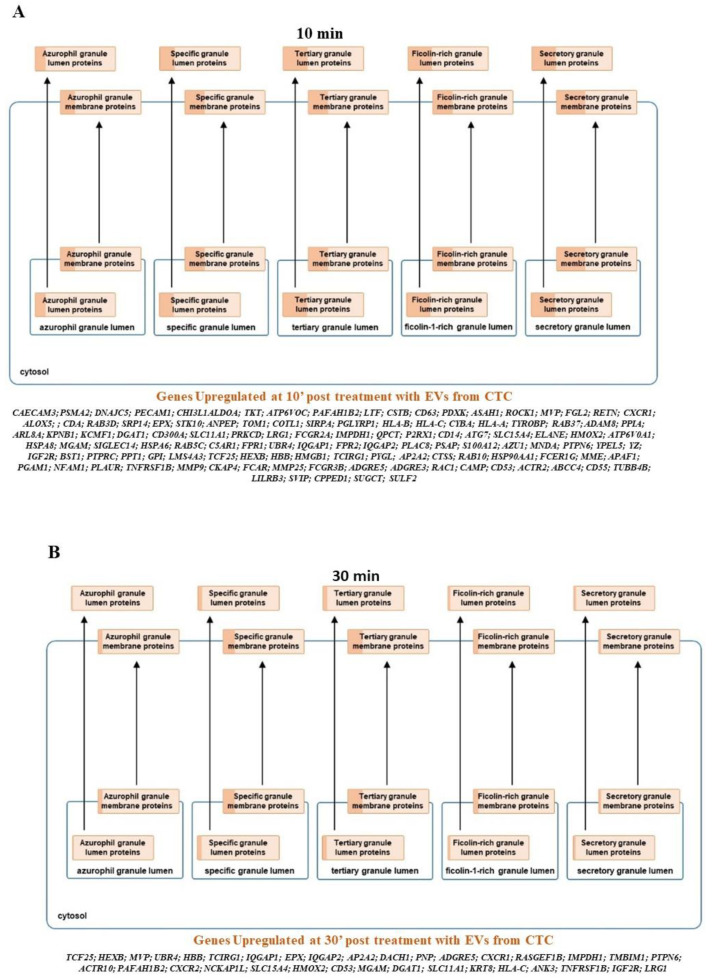
Summary of genes upregulated on exposure to EVs from CM61 that form part of the neutrophil degranulation pathway. The shading in the cellular component indicates the enrichment of number of genes in the said component with more of the granule being shaded indicative of more genes upregulated in that particular component. (**A**) Global change of proteins across all granular compartments are observed at 10 min There is no specific distribution of proteins across one of the cellular components and is a global change of all proteins at this time point; (**B**) 30 min, there is a complete reduction of proteins that constitute the azurophil and secretory granule lumens, while at (**C**) 60 min, the azurophil granule proteins are still downregulated while there is a slight increase in proteins that constitute the secretory granule lumen. The levels of specific, tertiary and ficolin1 rich granules do not show major changes between 30 min and 60 min.

**Figure 5 cancers-13-02727-f005:**
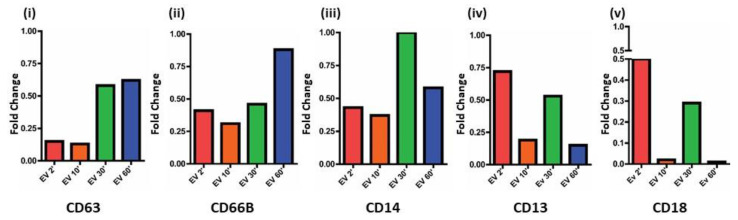
Comparative transcript level comparisons of neutrophil granule mobilization markers; (**i**) azurophil granules (CD63), (**ii**) specific granules (CD66b), and (**iii**) secretory vesicles (CD13, CD14, and CD18). With respect to time point 2′, there is a slight decrease in the levels of CD63, CD66b, CD14, (**iv**) CD13, and (**v**) CD18 followed by which there is an increase in all proteins at time point 30′; this is followed by CD63 and CD66b continuing to increase with time point 60′ while CD14, CD13, and CD18 show a sharp decline in transcript levels at 60′.

**Table 1 cancers-13-02727-t001:** Table of phosphorylated proteins identified by mass spectrometry in the EVs isolated from CTCs with active site identified and biological significance of the site in protein signaling pathways.

Protein	Phosphorylation Site(s)	Biological Significance
CRAF	S621	Essential for preventing protein degradation (Activation Site)
BRAF	S365; S729	14-3-3 Binding; BRAF-CRAF heterodimer formation (Activation Site)
MEK1	S222	Activation Site
AKT1	S122; T308	Activation Site
ERK2	T185; Y187	Activation Site
RSK2	S227; S369	Activation Site
YAP1	S109	Activation Site

**Table 2 cancers-13-02727-t002:** Table of phosphorylated proteins identified by mass spectrometry in the EVs isolated from CTCs with active site identified and biological significance of the site in protein signaling pathways.

Time Point	Differentially Regulated Genes	Family of Genes Upregulated (%)	Family of Genes Downregulated (%)
10′ EV treated vs. 2′ Naïve Untreated	Upregulated: 1676 Downregulated: 1234	Multicomplex: 62.53 Coding: 24.82 Non Coding: 8.89 Unassigned: 3.34 Pseudogenes: 0.42	Multicomplex: 88.01 Coding: 8.35 Non Coding: 3.08 Unassigned: 0.57
30′ EV treated vs. 2′ Naïve Untreated	Upregulated: 638 Downregulated: 511	Multicomplex: 86.36 Coding: 8.46 Non Coding: 4.08 Unassigned: 1.1	Multicomplex: 78.28 Coding: 8.41 Non Coding: 8.02 Unassigned: 5.28
60′ EV treated vs. 2′ Naïve Untreated	Upregulated: 428 Downregulated: 462	Multicomplex: 86.45 Coding: 9.58 Non Coding: 3.27 Unassigned: 0.7	Multicomplex: 77.49 Coding: 8.87 Non Coding: 7.58 Unassigned: 6.06

## Data Availability

The transcriptomics data discussed in this publication have been deposited in NCBI’s Gene Expression Omnibus and are accessible through GEP series accession number GSE175773. The mass spectrometry data have been deposited to the ProteomeXchange consortium via the PRIDE partner repository with the dataset identifier PXD026340.

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
