# Peer review of "Modulation of Early Neutrophil Granulation: The Circulating Tumor Cell-Extravesicular Connection in Pancreatic Ductal Adenocarcinoma"

_cancers, 2021, doi:10.3390/cancers13112727_

Round 1

Reviewer 1 Report

This article is based on data in the literature reporting that pancreatic cancer presenting cells dissociated from the primary site and enter into systemic circulation n (Circulating Tumor Cells, CTCs) ,  either alone or in clusters , have a greater predisposition towards forming distal metastases than single CTCs. The authors provide evidence for an extravesicular mode of communication between pancreatic cancer CTCs and neutrophils. This evidence is very interesting as it could add a further step to the understanding of how the circulating tumor cells communicate with other cells in circulation. The hypothesis of Mollinedo is also very interesting, which would therefore be in line with the above, postulating the possibility of “neutrophil-drawn carriages” in the process of cancer metastasis . Based on this model, neutrophils cluster around tumor cells protect them physically and transport them to different sites. The topic presented in the text is very interesting, of current interest. The article is very well written and is very clear and understandable.

Reviewer 2 Report

This is a well-written and well-presented manuscript investigating a novel phenomenon. It aims to understand the cargo of EVs derived from CTCs and to determine their effect on neutrophils and platelets. There are a few areas where further clarification or detail would improve the manuscript (in order of appearance):

line 25 - remove the additional 'how'

line 26 - the term NETosis should not be used in the simple summary and should be defined the first time it is used elsewhere, rather than just referencing

line 42 - shouldn't 'mortality' be 'survival'?

line 227 - it is important to provide some justification for this method of growing the CTCs.  In the discussion,  there should be comment on whether the source of CTCs used and the method of culture used is relevant to the real patient situation

line 229 - there is reference to the observation of a mixed population of cells for each line.  Was any attempt made to separate these or to understand whether one or both were contributing the EVs? A comment on this in the discussion would be valuable

line 409 - contents of this paragraph might be better in a table rather than text

line 554 - what is mean by 'aka'?

line 606 - I would prefer 'allow' to be changed to 'may allow'.  There is no evidence for this in the paper - the statement is a hypothesis (and a reasonable and interesting hypothesis) arising from the data

Reviewer 3 Report

In this manuscript the authors describe how the circulating tumor cells modulate neutrophil degranulation via extravesicular (EV) bodies. The authors use a range of high throughput techniques like mass spec and RNA-seq to characterize the EVs in depth. Further, the authors comment on the neutrophil degranulation and reteplase activity pathways.

This is an interesting study and may present value to the field. However, there are certain major concerns that I feel should be addressed to properly validate this study. They are listed below-

  1. Fig 1 A- size of EV should be quantified and presented as bar graphs with technical replicates.
  2. Fig 1B- Was equal amount of lysate loaded in western blots? If yes, then why are the CM61 protein markers so low expressed?
  3. Fig 1C- Please mention i and ii briefly in the figure also.
  4. Fig 1C- The validation of TSG 101 shRNA knockdown is not shown. A western blot showing effective KD must be shown.
  5. Fig 1C- Why was only CM61 cell line used for TSG101 KD and not the other cell lines?
  6. Fig 1C- The authors must also confirm the effect of TSG101 KD on EV markers by western blot as in Fig 1B.
  7. The authors could further validate the active signaling in EV cargo by using MAPK pathway inhibitors and further showing loss of the phosphorylated proteins in the cell line as well as the EVs. This will validate the observation.
  8. Fig 1E- The lanes are not marked in the figure. Are these duplicates? If so, it should be mentioned.
  9. Fig 2A- Why are 10’ exosome points not clustered? The authors must provide an explanation for this.
  10. Figures 2 and 3 are impossible to read. The authors must significantly improve the figure fonts and quality so that they are legible.
  11. Fig 2- Methods for RNA-seq analysis is completely missing. The authors must provide in detail which tools, algorithms and parameters etc. were used to generate the heatmaps and the supplementary data provided.
  12. In lines 409-423, the authors write a lot just stating the percentages. Tabular presentation will be appreciated here.
  13. Figure 3 presentation (Apart from increasing the fonts size) can be improved. The green shading to show a subset of genes is not effective and will confuse the reader.
  14. Figure 4- Error bars and statistics is missing. Are these technical replicates? Proper statistics should be presented on the graphs.
  15. Figure 4- Why is there fluctuation between CD14, CD13 and CD18? The authors must justify this.
  16. The central theme of the manuscript is around neutrophil degranulation, but the authors have not shown neutrophil degranulation by any assay.
  17. Reteplase activity can also be shown either by direct assays or by assaying any downstream targets.

Minor comments-

  1. There is too much literature review in the results sections. This can be made concise to include only relevant background.
  2. Some sections, like 3.5 do not cite any figures. May be re-organize the manuscript?
  3. The title can be rephrased to provide a clear message about the contents of the manuscript.

Round 2

Reviewer 3 Report

The authors have made considerable changes to the manuscript as suggested. While the authors have addressed most of the issues in the revised manuscript, some minor changes will make the manuscript even better.

  1. Font size of Figure 1B legend and Figure 4 can be increased to make sure that reader has no difficulty. Perhaps printing out the figure after preparing it may help?
  2. There are “?” signs in Figure 3A and S3A and S3B. Please make sure to edit these.
  3. Line 642- CXCL12-CXCL-4 pathway in lung injury part does not cite any reference?
  4. The question regarding using MAPK inhibitors raised in comment number 8 is not satisfactorily answered. The data presented by authors does not justify the question. It is understandable that mass spectrometry analysis requires more time. But the authors should at least include this in the discussion part where limitations of this study are discussed. This will add more value to the field and inform the audience about the limitations of Nanosight tracking analysis and the necessity of mass spectrometry for detection of phosphorylated proteins in the EVs.

Apart from these minor comments, the authors have done a satisfactory job in addressing the concerns. I congratulate the authors on their excellent work.
